# Myocardial Infarction in Children after COVID-19 and Risk Factors for Thrombosis

**DOI:** 10.3390/diagnostics12040884

**Published:** 2022-04-01

**Authors:** Eliza Cinteză, Cristiana Voicu, Cristina Filip, Mihnea Ioniță, Monica Popescu, Mihaela Bălgrădean, Alin Nicolescu, Hiyam Mahmoud

**Affiliations:** 1“Carol Davila” Pediatrics Department, University of Medicine and Pharmacy, 020021 Bucharest, Romania; mihaela.balgradean@umfcd.ro; 2“Marie Curie” Emergency Children’s Hospital, 041451 Bucharest, Romania; filipcristina06@yahoo.com (C.F.); mihnea-i@hotmail.com (M.I.); doc.monica.popescu@gmail.com (M.P.); nicolescu_a@yahoo.com (A.N.); hyam22@hotmail.com (H.M.); 3Royal Brompton Hospital, London SW3 6NP, UK

**Keywords:** COVID-19, myocardial infarction in children, Kawasaki disease, coronary aneurysms, thrombosis

## Abstract

Acute myocardial infarction (AMI) in children is rather anecdotic. However, following COVID-19, some conditions may develop which may favor thrombosis, myocardial infarction, and death. Such a condition is Kawasaki-like disease (K-lD). K-lD appears in children as a subgroup of the multisystem inflammatory syndrome (MIS-C). In some cases, K-lD patients may develop giant coronary aneurysms. The evolution and characteristics of coronary aneurysms from K-lD appear to be different from classical Kawasaki disease (KD) aneurysms. Differences include a lower percentage of aneurysm formation than in non-COVID-19 KD, a smaller number of giant forms, a tendency towards aneurysm regression, and fewer thrombotic events associated with AMI. We present here a review of the literature on the thrombotic risks of post-COVID-19 coronary aneurysms, starting from a unique clinical case of a 2-year-old boy who developed multiple coronary aneurysms, followed by AMI. In dehydration conditions, 6 months after COVID-19, the boy developed anterior descending artery occlusion and a slow favorable outcome of the AMI after thrombolysis. This review establishes severity criteria and risk factors that predispose to thrombosis and AMI in post-COVID-19 patients. These may include dehydration, thrombophilia, congenital malformations, chronic inflammatory conditions, chronic kidney impairment, acute cardiac failure, and others. All these possible complications should be monitored during acute illness. Ischemic heart disease prevalence in children may increase in the post-COVID-19 era, due to an association between coronary aneurysm formation, thrombophilia, and other risk factors whose presence will make a difference in long-term prognosis.

## 1. Introduction

Caused by the severe acute respiratory syndrome coronavirus 2 (SARS-CoV-2), COVID-19 is an ongoing pandemic that has resulted in over 412 million confirmed cases and over 5.8 million deaths globally [1]. Although only 1% of the children under 10 years old infected with SARS CoV-2 virus develop COVID-19, with the Delta and Omicron variants, admissions for children increased [2,3]. COVID-19 initially manifested itself in fewer, milder, and better prognosis forms of the disease in children compared to adults [3]. Although the main damage is respiratory, the involvement of the cardiovascular system substantially changes the prognosis [3].

Cardiovascular disease associated with COVID-19 infection usually falls into two categories—first, myocardial damage from the initial infection, following the respiratory disease and, second, subsequent damage from Multisystemic Inflammatory Syndrome in Children (MIS-C), in less than 1% of the infected children [4]. Cardiovascular complications of severe SARS-CoV-2 infection are centered around cases of myopericarditis, fulminant myocarditis, pulmonary hypertension, or heart rhythm disorders [3,4,5,6,7]. Severe cases have also been reported in patients with known congenital heart disease, putting them in a fragile balance [3].

MIS-C occurs between 2 and 6 weeks after the acute onset of SARS-CoV-2 infection, sometimes in a previously asymptomatic patient, and consists of a systemic hyperinflammatory status having symptoms similar to septic shock or Kawasaki disease. This syndrome was described for the first time by the Royal College of Paediatrics and Child Health as a pediatric multisystem inflammatory syndrome (PMIS) [8]. It is assumed to be a delayed immune response to COVID-19, which can lead to severe cardiovascular involvement [9]. The Centers for Disease Control and Prevention and the World Health Organization then labeled the condition as a multisystem inflammatory syndrome in children (MIS-C) with very well-established diagnostic criteria (Table 1) [10]. Complications due to MIS-C are more severe, due either to the appearance of severe forms of shock, initially hypovolemic, then cardiogenic, or to the appearance of giant coronary aneurysms, with evolution to intravascular thrombosis and myocardial infarction. The death rate in MIS-C patients is 2% [3,11,12,13,14].

Myocardial infarction in children is extremely rare, with most suspected cases being in fact cases of fulminant myocarditis. Myocardial infarction is secondary to acute coronary thrombosis. The predisposition for the appearance of a thrombus depends on several factors: alteration of the vascular endothelium, modification of blood coagulation parameters by the appearance of thrombophilia, and hemodynamic changes of the blood vessel (stasis, turbulence). These conditions are fulfilled point by point in MIS-C with aneurysmal coronary dilation, similar to those that appear in children with classical Kawasaki disease.

Kawasaki disease (KD) is a medium-sized systemic vasculitis that affects, with predilection, coronary arteries, predominantly in children <5 years of age. Although the etiology of KD is not clearly established, it is generally accepted that viral agents can trigger the disease. It has been postulated that seasonal peaks of the disease coincide with the seasonality of common respiratory infections. Coronavirus is implicated in this epidemic pattern [15,16,17]. Because the dilated coronary arteries change the rheology of blood, the risk of ischemic events is increased as a result of sluggish or turbulent blood flow [18,19,20]. The risk for cardiac coronary sequelae after Kawasaki disease is 2.6% at 30 days after the onset of the disease. About 9% of patients develop cardiac complications in the acute and subacute phases of the disease, even if they receive the correct treatment at an optimal time [21]. Predictive factors for a progressive evolution that may appear in 4.2% despite correct treatment are larger coronary artery lesions, age ≥60 months, recurrent status, or parental history of KD [22]. The highest risk for acute coronary syndrome in children is related to giant aneurysm formations of more than 8 mm in diameter. The prevalence of giant coronary aneurysms after KD is estimated at 0.16% [23]. During the COVID-19 pandemic, surprisingly, the number of classical Kawasaki diseases decreased significantly, suggesting the possibility of a pathogen-related transmission [24].

Herein, we will provide a systematic overview of a previously healthy child who had a major cardiovascular complication after COVID-19 infection. He developed giant coronary aneurysms after MIS-C, Kawasaki-like type, which, despite medium-term cardiac monitoring and therapy, ended with acute myocardial infarction. This is a unique case report in the medical literature of an extended myocardial infarction on the background of multiple giant coronary aneurysms complicated with thrombosis, 5 months from initial MIS-C diagnosis.

## 2. Case Report

We present the case of a 2-year-3-month-old boy with a significant medical history of COVID-19 infection followed by paediatric multisystem inflammatory syndrome (PMIS). At 1 year and 9 months old, the child entered the community, but after only 2 weeks he developed respiratory symptoms, cough, and fever, he was treated at home with symptomatic medication and antibiotics for 5 days, which led to the complete resolution of symptoms. Without having any evidence of COVID-19, we considered this episode a ‘terminus a quo’ because our patient developed signs and symptoms of PMIS 4 weeks after the SARS-CoV2 onset infection, a diagnosis confirmed by the laboratory tests. Four months after the PMIS episode, this patient presented in our clinic with ST-Elevation myocardial infarction. He was born at term, after an uneventful pregnancy (gestational age 36 weeks, birthweight 2800 g, APGAR score 10). Both parents were in good health, with no medical history of chronic diseases. The patient was admitted into our clinic, transferred from a regional hospital with suspicion of acute myocardial infarction. The acute disease started three days before presentation, with diarrhea (4 stools/day) and vomiting (1 episode/day), over which he had a paroxysmal crisis manifested with generalized hypotonia, cyanosis, and loss of consciousness for about 20 min. After clinical evaluation, ECG, and echocardiography, he was transferred to our hospital for an interventional diagnostic and therapeutic approach.

Past medical history revealed an acute pharyngitis episode at the age of 5 months, COVID-19 infection complicated with Kawasaki-like PMIS at the age of 1 year and 9 months (fever, diarrhea, anorexia, vomiting, facial erythema), and mixed-agglutinin autoimmune hemolytic anemia post-COVID-19. The paediatric multisystem inflammatory syndrome was diagnosed after clinical, biological, and echocardiographic evaluation. He presented high persistent fever, skin rash (erythematous eruption on the trunk and limbs and macular perioral erythema), and gastrointestinal manifestations (diarrhea, vomiting, anorexia). Biological investigations showed non-specific inflammatory syndrome, increased values of D-Dimers and NT-proNPB, as well as positive serologic testing for IgG antibodies against SARS-CoV-2. Echocardiography revealed circumferential pericardial and pleural effusion, left coronary artery dilation of 3.4 mm (Z score + 3.8). A severe mixed agglutinin-mediated autoimmune hemolytic anemia related to COVID-19 infection was diagnosed and he received iso-group, iso-Rh transfusion therapy following treatment with iv immunoglobulins (IVIG) (IgVena), corticosteroid therapy (Solumedrol), rituximab (Mabthera), antiaggregant (Aspenter), subcutaneous anticoagulant (Clexane), and antibiotic and symptomatic therapies. The evolution was favorable, with discharge after one month and recommendations for antiaggregant therapy and subsequent pediatric cardiologist follow-up.

Clinical examination at presentation in our hospital revealed an afebrile 2-year-old male (36.7 °C, axillary) with a height of 90 cm, a weight of 14 kg, relatively good general condition, pale, dehydrated, respiratory rate of 25/min, oxygen saturation at 98% in room air, systolic murmur II/VI, protodiastolic third sound, BP at 110/80 mmHg, HR at 127/min, diuresis present, and normal consciousness.

Complete Blood Count revealed 4.930 WBC/mm^3^, Lymphocytes 34.5% (1.700/mm^3^), Neutrophils 57.2% (2.820/mm^3^), Monocytes 8.3% (410/mm^3^), RBC 4.560.000/mcL, Hb 12.2 g/dL, Hct 34.6%, and Platelets 430.000/mm^3^. Serum electrolytes were normal, except mild hypokalemia (Na^+^ 137.2 mmol/L, K^+^ 3.36 mmol/L, Ca^2+^ 1.26 mmol/L), with normal serum lactate (0.28 mmol/L) and CK: 2563.00 IU/L, CK-MB: 463.60 IU/L (NV < 24), Troponin T > 2.000 ng/mL (NV < 14), TGO: 288.8 IU/L, TGP: 35.3 IU/L, urea: 11.4 mg/dL, and creatinine: 0.24 mg/dL. Follow the decrement of the cardiac enzymes in Table 2.

Electrocardiogram (ECG) showed sinus rhythm, QRS axis +90, HR 120/min, pathologic Q wave, DI, aVL, 3 mm ST-segment elevation in leads V1–V6, DI, and aVL, reduced/absent R wave in V2–V6 (Figure 1).

The echocardiographic assessment showed dyskinesia of the left ventricular apex and akinesia of the apical portion of the anterior and lateral wall with an estimated left ventricular ejection fraction (LVEF) around 35%. The coronary artery dilation was visible at the level of the left main artery, right coronary artery, and an aneurysmal dilation of the anterior descending artery (ADA) (Figure 2, Figure 3 and Figure 4).

Emergency coronary angiography (Figure 5) revealed dilation of the left main artery (LMA 2.9 mm, z score +2.2), aneurysmal dilation of the ADA measuring ~3.8 mm, z score +6.3, and thrombosis in the middle part of the aneurysm and stop flow. The origin of the circumflex artery (LCX) measured 2.6 mm in diameter (z score +2.8) and was followed by aneurysmal dilation of 6.2 × 19 mm (z score +12.3) with a non-obstructive thrombus inside the aneurysm. LCX continues filiform but with other distal dilations. The right coronary artery was permeable, with good flow to distality with a diameter of 4 mm at the origin (z score +5.6).

We established the diagnosis of an acute anterior and lateral ST-elevation myocardial infarction due to thrombosis of the ADA and LCX arteries, accompanied by multiple aneurysmal coronary artery dilation. Because it was impossible to perform percutaneous coronary intervention (giant multiple aneurysms, small patient size, aspiration of thrombus not recommended, level of evidence A, Class III), [25] we initiated iv thrombolytic therapy with tissue plasminogen activator (tPA: Alteplase 0.05 mg/kg/h) for 12 h, inotropic support with Noradrenaline (0.02 μg/kg/min), and continuous infusion with Heparin in a separate iv line (10 UI/kg/h). Every 4 h, we took a blood sample to monitor: fibrinogen (in order to maintain fibrinogen >150 mg/dL to avoid bleeding complications), aPTT, troponin T, CK, CK-MB, ATIII (to maintain a level of 80–120% of normal activity), and transaminase levels (Table 2).

Once the tPA was stopped, we increased the level of Heparin to achieve the target activated clotting time of 200–250 s and introduced an inhibitor of platelet activation and aggregation, Clopidogrel (1 mg/kg/day). Subsequently, Heparin was replaced by subcutaneous anticoagulant medication with Enoxaparin (100 UI/kg/dose, twice daily) and after 14 days, with acenocoumarol (Sintrom). In order to improve cardiac function, we also adjusted the medication by introducing an ACE inhibitor, lisinopril (0.1 mg/kg/day), a beta-blocker, bisoprolol (0.1 mg/kg/day), and spironolactone, an anti-aldosterone diuretic.

Serial monitoring of dynamic ECG changes showed progressive remission of ST-segment elevation (Figure 6), dynamic T-wave inversions, and pathological Q waves.

Subsequent echocardiographic evaluations demonstrated a slow enhancement of the cardiac function, with maintenance of the dyskinetic movement of the LV apex. We couldn’t achieve the complete resolution of the thrombus initially. An attached mobile thrombus was visible for a couple of days, with interesting imaging revealing a secondary, larger aneurysm of the ADA of 10 × 8 mm, z score +24 after reperfusion and opening of the first one with an acceptable blood flow at the level of ADA and LCX.

During his hospital stay, the patient experienced one dermatological complication with uncertain etiology, two zones of alopecia areata (Figure 7). Alopecia areata appeared either due to stress exposure, as an adverse effect to ACE inhibitor, or triggered by coronary artery abnormalities [26,27,28,29]. We changed the lisinopril (ACE inhibitor) to losartan (angiotensin receptor blocker) and associated a local treatment according to the dermatologist’s recommendation, with a favorable local evolution.

Subsequent evaluations after discharge (at 1 month) revealed the presence of the thrombus inside the second aneurysm of the ADA and increased CK-MB (37 U/L, NV < 24) and troponin T levels (27.8 pg/mL, NV < 14). We decided to associate double antiaggregant therapy (clopidogrel, aspirin) with anticoagulant (acenocoumarol) and the thrombus disappeared completely after 3 months, with improvement of the cardiac function parameters (EF increased to 45%).

## 3. Discussion

Myocardial infarction in children is a very rare entity, being mostly associated with Kawasaki disease or an anomalous origin of the coronary artery from the pulmonary artery (ALCAPA) [30]. During the time of the COVID-19 pandemic, numerous cases of Kawasaki-like diseasehave appeared, many of them associating aneurysmatic dilations of the coronary artery after evolving as a multisystem inflammatory syndrome in children (MIS-C). MIS-C has an incidence of 0.2–0.6% of all pediatric SARS-CoV-2 infections. Although mortality in MIS-C is limited to 2% of patients, the morbidity is high. Of MIS-C patients, 53% presented with shock, 52% with myocardial dysfunction, 27% with ECG changes, and 15% with coronary dilations. Of these patients, 75% needed admission to PICU and 4% necessitate extracorporeal membrane oxygenation [3]. The mortality of patients with K-lD associated with COVID-19 is 6.7% due to severe heart failure, cardiac arrest, and refractory hypotension. In these cases, no myocardial infarction was mentioned as a cause for death, but only brain infarction [31]. As can be seen from the same meta-analysis of over 688 patients with MIS-C, there are important differentiating features in the long-term prognosis between Kawasaki disease and MIS-C patients. The MIS-C patients have a higher risk of KD shock syndrome (50–60% vs. 2–7%), higher percentages for myocardial dysfunction (52% vs. <1%), a higher coagulopathies rate, a higher IVIG resistance rate (50–60% vs. 10–20%), higher long-term cardiac sequelae rate (5.5% vs. <5%), and doubled risk of exitus (2% vs. 1%), although the risk for coronary anomalies is higher for Kawasaki disease compared to MIS-C (25% vs. 15%) [3]. A comparative view of the most important published papers regarding MIS-C and cardiac complications is presented in Table 3.

Other differences between Kawasaki-like disease and classical Kawasaki disease were appreciated by Schranz et al. and are related to the primarily affected tissular structure. In the opinion of the authors, the microvascular endothelial cells are affected in the Kawasaki-like disease, compared to classic Kawasaki syndrome, which affects the epicardial coronary arteries. That is why the classical form of KD rarely coexists with myocarditis [36]. The risk of coronary artery formation in KD is 9% if the disease is correctly treated and 25% if treatment is delayed [3,21]. In the following 30 days, coronary aneurysms may persist in 4.2% of cases [22] and become a sequela in 2.6% of cases over that period [21]. Of this percentage, 0.16% are giant coronary aneurysms, which necessitate anticoagulation [23]. Compared with coronary aneurysm results after COVID-19 evolution with MIS-C, 15% of the patients with MIS-C will develop a coronary aneurysm, but all of them will resolve. Sequelae are not being reported, 0% being found at echocardiographic evaluation. Our case is an exception to this favorable evolution.

Our patient went through multisystem inflammation associated with COVID-19, coming out with multiple giant coronary aneurysms which developed thrombosis and myocardial infarction at a 5-month interval from the MIS-C appearance. We can link this dramatic evolution of our 2-year-old patient to the COVID-19 pandemic, as the serology test for SARS-CoV-2 antibodies was positive from the beginning. In this background of coronary arteries dilatations refractory to treatment, he suffered from an acute myocardial infarction in a condition of dehydration, being on antiaggregant treatment (according to current guidelines, the ideal treatment should have included an anticoagulant) [11]. Myocardial infarction in children is a very rare occurrence. It is well known that is the main cause of death in patients with KD as a result of thrombotic occlusion in a coronary aneurysm. In a study performed by Celermeyer et al. on 17 patients, the symptoms were dyspnea, vomiting, and anorexia (almost similar to our patient). The same information resulted also from a large study by Kato et al., conducted on 195 patients. They concluded that myocardial infarction usually occurred within the first year of illness, had clinical manifestations of shock, crying, vomiting, abdominal pain, and only in children older than 4 years, chest pain. One-third of patients did not have any symptoms in the Kato et al. study. In 63% of the patients, it occurred during sleep or at rest, similar to our case [37]. The fatality rate after the first attack was 22%, but 60% remained with subsequent cardiac dysfunction and left ventricle aneurysm formation, as in our case. The mortality after myocardial infarction in the study performed by Celermeyer in children was 47% with a low incidence of late cardiac arrhythmias in survivors and a good effort tolerance independent of the ejection fraction [30]. Recently, Tsuda et al. did a retrospective analysis of 60 patients who experienced MI between 1976 and 2007 and concluded that one of five patients subsequently died (20%) [38].

Increased risk of thrombotic complications usually comes from coronary artery aneurysms, ventricular dysfunction, hypercoagulable state, endothelial lesion, and vascular stasis due to immobilization [3]. Patients with preexisting cardiac problems (genetic disease, congenital heart disease with a significant shunt, obstruction, or cyanosis), giant coronary aneurysm, heart failure, ventricular arrhythmias, pulmonary hypertension, end-stage patients, and patients with a procoagulant status or thrombophilia should avoid or immediately treat any situation that might precipitate toward an unpredictable evolution. Those situations are identified with fever, dehydration, electrolyte imbalance, and adrenergic stress [3]. According to studies, the most important risk factor for thrombosis is not the SARS-CoV-2 infection itself, but the proinflammatory and procoagulant state reported in MIS-C cases [39]. Modifications in procoagulant status evaluation in COVID-19 or in MIS-C patients consist of increasing D-dimer concentration, a mild to moderate decrease in thrombocytes count, and a prolongation of the prothrombin time. Similar changes may occur in sepsis, but of varying intensity, associating severe thrombocytopenia and mild damage to the D-dimers. As an expression of thrombotic microangiopathy, lactate dehydrogenase (LDH) and high ferritin levels may be present [40].

Patients with a persistent giant coronary aneurysm as a cardiac sequela after classical Kawasaki disease or after Kawasaki-like disease after MIS-C are extremely fragile. For them, the coronary aneurysm represents the risk of an acute coronary event, even in children, and at this age, the probability for an acute coronary syndrome is very small. Any additional factor that will increase the blood viscosity or generate a hypercoagulable state or thrombophilia must be avoided (Table 4). Important recommendations should be addressed in any situation associated with dehydration, electrolyte imbalance, hypercoagulable state, increased viscosity, thrombophilia, or thrombocytosis [3].

Guidelines recommend that the management of a child who had AMI should be the same as for adults, despite the differing etiology [41]. In the evolution of our patient, signs of myocardial ischemia diminished rapidly after tPA infusion, as shown by electrocardiogram (Figure 6) and echocardiography. The next few days after thrombolysis, the cardiac contractility improved with an estimated LVEF of 45% (Simpson’s method) and the thrombus decreased in size, but without complete resolution at discharge. He was transferred to the cardiology ward after 3 days of admission in the pediatric intensive care unit, with a good clinical condition. After a long time, monitoring, and dose adjustment of the treatment (24 days), he was discharged and ambulatory followed-up every 2 to 4 weeks (Figure 8). Although the giant coronary aneurysms persist, the patient’s general condition is normal under double antiaggregant therapy associated with anticoagulant treatment.

Our patient presented with acute myocardial infarction, being previously diagnosed with coronary artery dilation on both right and left coronary arteries, but without the mention of a giant aneurysm. Aneurysms were not present at the origin of the vessels, only on the anterior descending artery and on the circumflex artery into the middle part of both vessels. Thus, echocardiography might have missed them, taking into consideration the difficulty for evaluation of all the coronary segments by echocardiography. Previously to the patient’s presentation in our hospital with AMI, he was evaluated by at least three pediatric cardiologists and none suggested the need for a more detailed imagistic evaluation of coronary anatomy. Echocardiography seems to be insufficient to rely on when hidden coronary aneurysms are suspected.

## 4. Conclusions

With this case as an example of an unfavorable evolution of COVID-19 complicated with Kawasaki-like paediatric multisystem inflammatory syndrome, we strongly recommend that superior imagistic methods (angioCT, angioMRI, coronary angiography) be recommended in patients with COVID-19 and MIS-C in their past medical history, with Kawasaki-type clinical manifestations and temporary dilation of the coronary arteries, which disappear after the acute phase or with permanent dilations in the proximal segments, in order to exclude the hidden giant aneurysm.

An evaluation for the procoagulant state should be completed for each patient that has a predisposition for coronary vasculopathy. Therefore, patients at risk should avoid situations associated with dehydration, increased viscosity of the blood, and electrolyte imbalance and administer prophylactic treatment if recommended.

Regarding patients after COVID-19 and MIS-C which associate coronary aneurysm, a preventive treatment of anticoagulant drugs or an antiaggregant treatment should be recommended according to the dimensions of the coronary artery. These patients, especially small children and infants, should be evaluated at short intervals in order to recommend proper medication according to the percentile score or Z score.

## Figures and Tables

**Figure 1 diagnostics-12-00884-f001:**
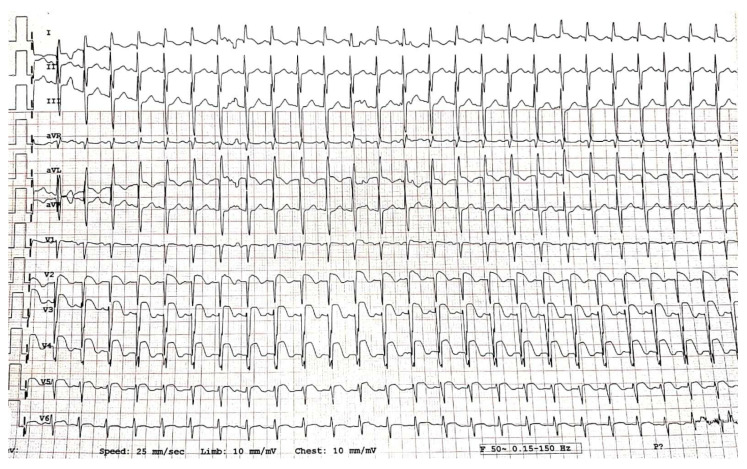
Electrocardiogram at admission.

**Figure 2 diagnostics-12-00884-f002:**
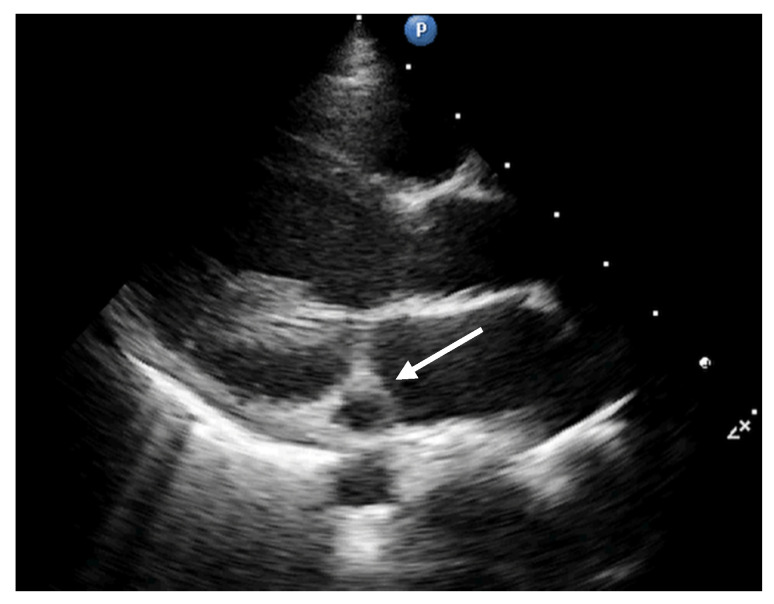
Parasternal long axis. Dilation of a vascular structure into the atrioventricular groove (circumflex artery: white arrow).

**Figure 3 diagnostics-12-00884-f003:**
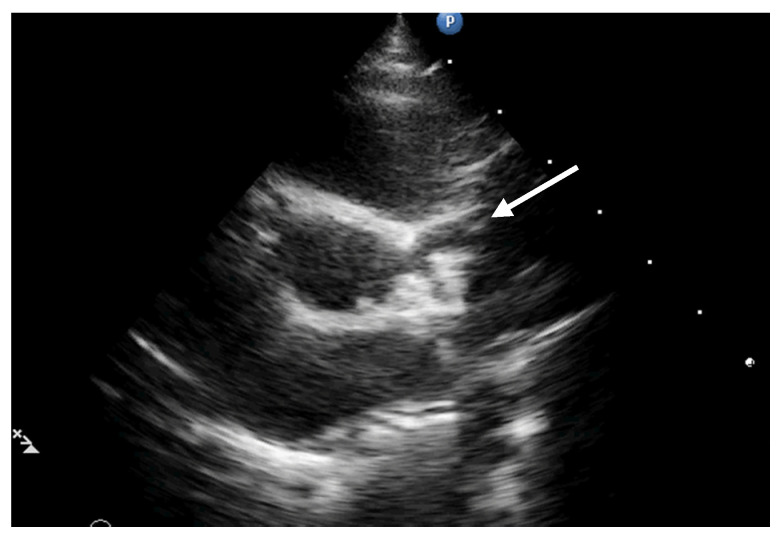
Parasternal short axis. Origins of the coronary arteries with dilation of the anterior descendent artery (white arrow).

**Figure 4 diagnostics-12-00884-f004:**
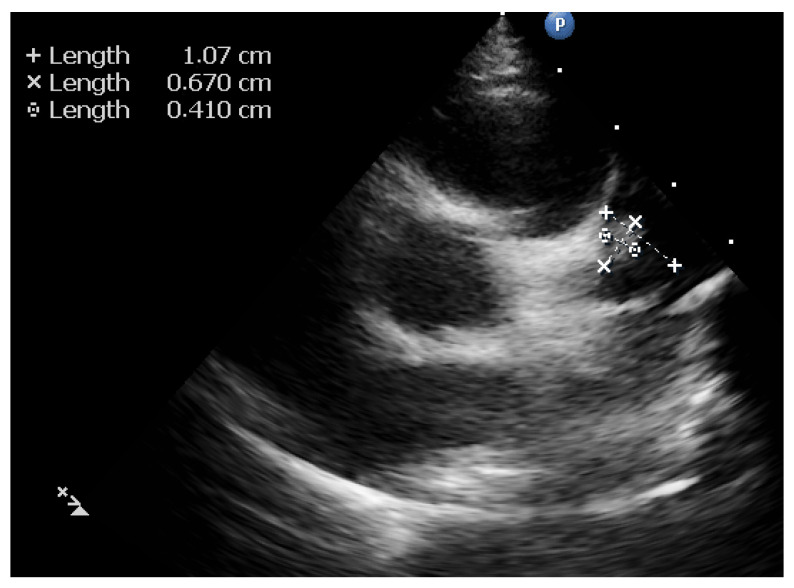
Parasternal Short axis. View of the aneurysm of the anterior descending artery with thrombus inside.

**Figure 5 diagnostics-12-00884-f005:**
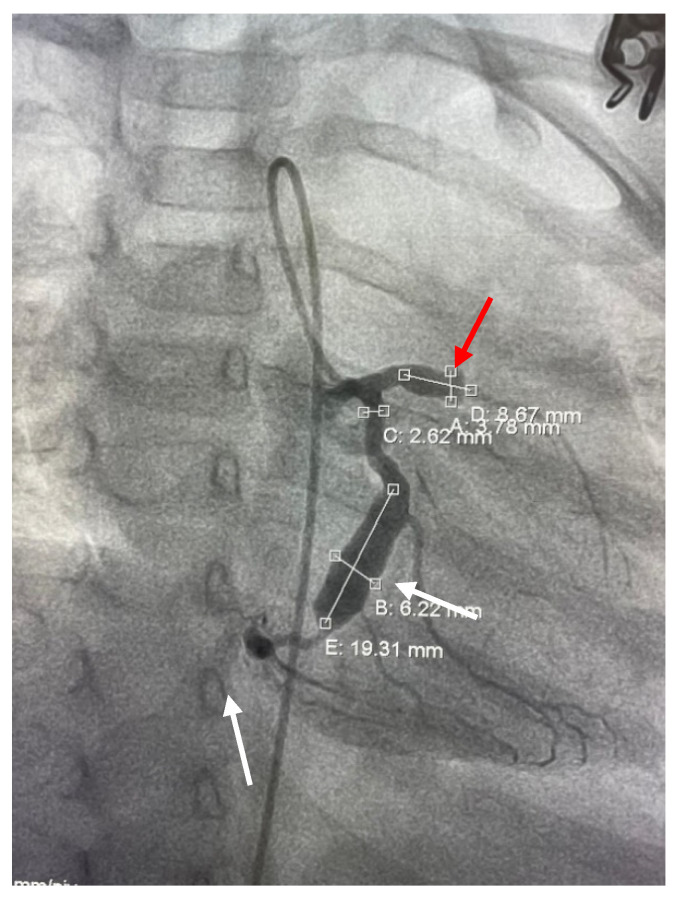
Coronary angiography. Multiple coronary aneurysms on the anterior descending artery (stop flow, thrombosis process in progress: red arrow) and on the circumflex artery (white arrows).

**Figure 6 diagnostics-12-00884-f006:**
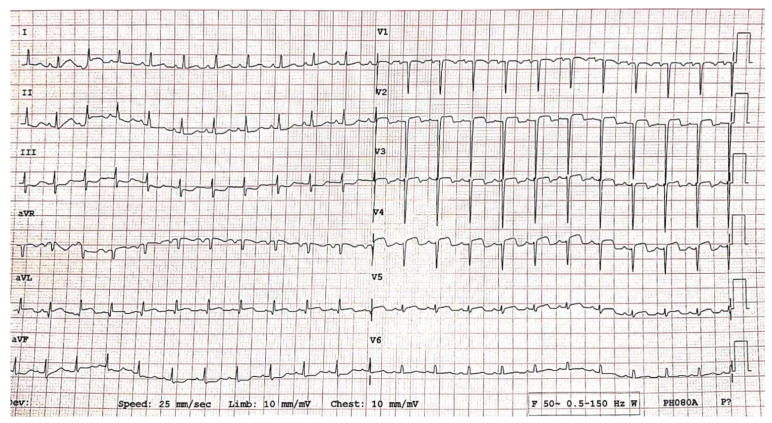
Electrocardiogram after thrombolysis, criteria of reperfusion.

**Figure 7 diagnostics-12-00884-f007:**
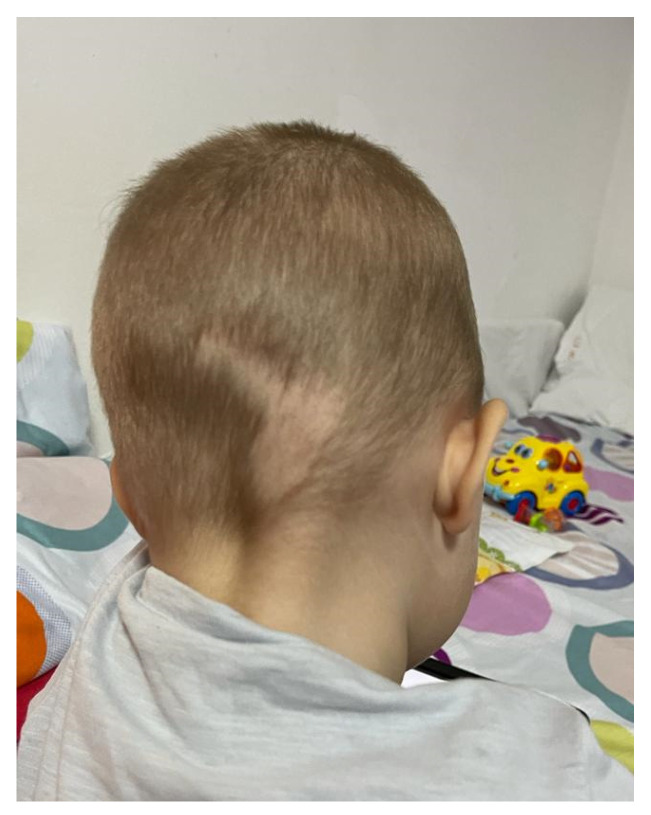
Alopecia areata, a complication during myocardial infarction recovery possibly related to the ACE inhibitors treatment.

**Figure 8 diagnostics-12-00884-f008:**
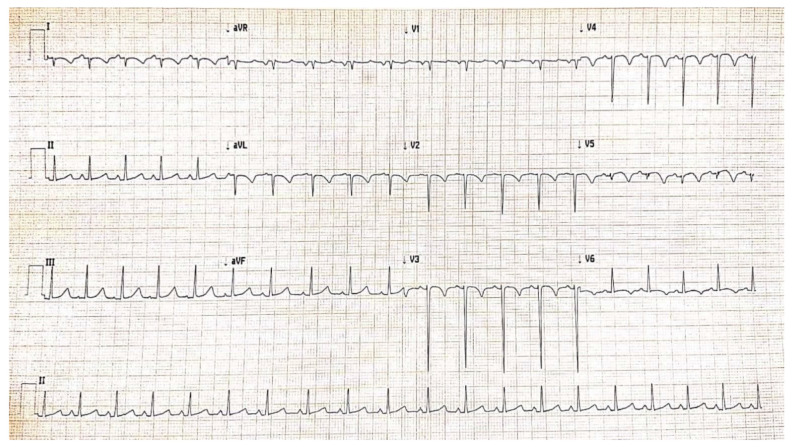
Electrocardiogram at discharge.

**Table 1 diagnostics-12-00884-t001:** Comparative view of the diagnostic criteria of Kawasaki disease, incomplete or atypical Kawasaki disease, and pediatric multisystemic inflammatory syndrome, with permission from Voicu et al. [9,10,11]. COVID-19: coronavirus 2019 disease; LAD: left anterior descending; NTproBNP: N-terminal pro Brain type natriuretic peptide; RCA: right coronary artery; RT PCR: reverse transcriptase-polymerase chain reaction; WBC: white blood cells.

Kawasaki Disease (KD)	Incomplete (or Atypical) KD	Pediatric Multisystemic Inflammatory Syndrome (Required all 6 Criteria)
Fever, and 4/5 criteria: - Erythema and cracked lips, strawberry tongue, and/or erythema of the pharynx and oral mucosa - Bilateral bulbar conjunctival injection - Rash maculopapular, erythematous - Erythema and edema of the hands and feet in the acute phase or periungual desquamation in the subacute phase - Cervical lymph nodes ≥ 1.5 cm.	- Children with: · Prolonged Fever (≥ 5 days) · 2–3 criteria OR - Infants with Prolonged Fever (≥7 days without other explanation) - Compatible laboratory tests (3 of the 6 criteria) o anemia o thrombocytosis after the 7th day of fever o albumin level ≤3 g/dL o elevated ALT level o WBC ≥ 15,000/mm^3^ o urine ≥ 10 WBC/hpf - Compatible echocardiographic findings (any of the following) o Z score LAD or RCA ≥2.5 o Coronary artery aneurysm o ≥3 features from: - Decreased LV function - Pericardial effusion - Z score LAD 2–2.5 - Mitral regurgitation	- Child 0–19 years - Fever ≥ 3 days - Clinical signs of multisystem involvement (at least 2 of the following): o rash/bilateral non-purulent conjunctivitis/mucocutaneous inflammation signs: oral, hands, or feet o hypotension or shock o features of myocardial dysfunction, pericarditis, valvulitis, coronary abnormalities (echo findings or troponin/NT proBNP) o evidence of coagulopathy (prolonged prothrombin time, partial thromboplastin time, or elevated D-dimers) o acute gastrointestinal symptoms (diarrhea, vomiting, abdominal pain) - Elevated markers of inflammation such as C reactive protein, procalcitonin, erythrocyte sedimentation rate. - No other obvious microbial cause of inflammation, including bacterial sepsis, staphylococcal/streptococcal toxic shock syndrome - Evidence of COVID-19 (RT PCR, antigen test, serology) or likely contact with patients with COVID-19

**Table 2 diagnostics-12-00884-t002:** Laboratory results (tPA: tissue plasminogen activator).

	Initial	4 h after Starting tPA	8 h Post-tPA	12 h Post-tPA	24 h Post-tPA	7 Days after Admission	At Discharge
Troponin T (ng/mL)	>2000	>2000	>2000	>2000	>2000	611	<40
Fibrinogen (mg/dL)	270	232	266	249	270	normal	normal
CK (IU/L)	2.563	1.937	1654	1446	1437	34	72
CK-MB (IU/L)	463	338	221	174	140	20	34.8
NT-proNBP (pg/mL)	7.857	-	-	7.078	-	7.837	8.030
TGO (IU/L)	288.8	258	234	210	161	normal	normal

**Table 3 diagnostics-12-00884-t003:** Comparative view of the most important published papers regarding MIS-C and cardiac complications [3,13,32,33,34,35].

Author	Feldstein	Verdoni	Whittaker	Grimaud	Moraleda	Belhadjer
**Country**	USA	Italy	UK	France	Spain	Switzerland/France
**Number of Patients**	186	10	58	20	31	35
**Age**	8.3	7.5	9	10	7.6	10
**Shock**	NR	5 (50%)	27 (46%)	20 (100%)	15 (48%)	28(80%)
**Myocardial Dysfunction**	70 (38%)	5 (50%)	18 (31%)	20 (100%)	15 (48%)	35 (100%)
**Coronary Artery Involvement**	15 (8%)	2 (20%)	8 (14%)	0 (0%)	3 (10%)	6 (17%)
**Coronary Sequelae**	0 (0%)	0 (0%)	0 (0%)	0 (0%)	0 (0%)	0 (0%)
**Death**	4 (2%)	0 (0%)	1 (2%)	0 (0%)	1 (3%)	0 (0%)

**Table 4 diagnostics-12-00884-t004:** Risk factors for an acute coronary syndrome in children.

1.	**Factors That Increase the Vascular Turbulence in Coronary Arteries** Coronary aneurysms (the risk increases with the diameter of the aneurysm) Coronary dilation after Kawasaki or Kawasaki-like disease Coronary stenosis After cardiac surgery, including the origin of the coronary arteries Thromboembolic events (embolization into a coronary artery) in endocarditis Coronary wall alterations secondary to chronic kidney disease
2.	**Factors Associated with Reduced Myocyte Supply** Heart Failure Hypotension Shock Systolic dysfunction Severe hypovolemia Severe anemia Severe hypoglycemia Severe cyanotic disease (oxygen saturation < 70–80%) Severe left heart obstructive disease-aortic stenosis, mitral stenosis, hypertrophic obstructive cardiomyopathy
3.	**Primary Factors That Increase the Risk for Thrombosis** Mutation of the Factor V Leiden Mutation of the prothrombin G20210A Mutation of factor XIII Deficiency in Protein C Deficiency in Protein S Deficiency in Antithrombin III Familial dysfibrinogenemia Congenital deficiency of plasminogen MTHFR mutation (homocysteine increased levels)
4.	**Secondary Factors That Increase the Risk for Thrombosis** Autoimmune disease-antiphospholipid syndrome, lupus anticoagulant, anticardiolipin antibodies, anti-beta 1 glycoprotein antibodies. Heparin-induced thrombocytopenia Thrombotic thrombocytopenic purpura Hemolytic-uremic syndrome Paroxysmal nocturnal hemoglobinuria COVID-19 Sepsis Sickle-cell disease Myeloproliferative disorders Essential thrombocytosis Policitemia vera Cancer Central venous catheter Nephrotic syndrome Membranous nephropathy Inflammatory bowel disease (ulcerative colitis, Crohn’s disease) Pregnancy Estrogen pills Obesity Sedentarism

## Data Availability

Not applicable.

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
