# Peer review of "Myocardial Infarction in Children after COVID-19 and Risk Factors for Thrombosis"

_diagnostics, 2022, doi:10.3390/diagnostics12040884_

Round 1

Reviewer 1 Report

Dear Authors,

This paper is absolutely original, authors after case report they present a narrative review in the discussion: I reccomend to explain better articles selection

Overall judgment cases report are well described, the topic is original and pictures and data reported are very interesting 

Reviewer 2 Report

Review of the Case report:

diagnostics-1626508

Myocardial infarction in children after COVID-19 and risk factors for thrombosis.

In this case report, the authors evaluate the thrombotic risk of coronary aneurysms provoked by COVID-19 infection in a two-year-old boy who posteriorly had an acute myocardial infarction (AMI).

The report is very well written and very interesting. The authors tried to evaluate every angle of the situation, achieving a very complete set of procedures that demonstrated the reported differences with the Kawasaki disease aneurysms.

Issues:

Perhaps I missed it in your article, if so excuse me, but it has been reported that the longer the infection is untreated or not well treated, the higher the probability of developing the multisystem inflammatory syndrome (MIS-C) ( here are some articles related to this: https://pubmed.ncbi.nlm.nih.gov/34703656/, https://pubmed.ncbi.nlm.nih.gov/33633944/ )

In general terms, what I am asking is, Were you able establish the number of days of infection prior to hospitalization?

In other words, it is important to report a clearer infection pattern and its relationship with time. As you know, MIS-C occurs between 2 -6 weeks after SARS-CoV-2 onset infection and it is not clear if the case was still infected by the time he was hospitalized.

Your following statement is in concordance to what I am requesting:

“We present the case of a 2-year-old boy with a significant medical history of COVID-19 infection followed by paediatric multisystem inflammatory syndrome (PMIS) who presented in our clinic with ST Elevation myocardial infarction”.

This statement needs to be clearer, what does “a significant medical history of COVID-19” means?

Please be more specific, for instance,

1) If the case was infected by the time of hospitalization,

2) Was the case infected more than once?

3) How long ago was the case infected?

4) How long does the infection or infections last?

This will help build a clearer study case.
